# Binding of κ-Conotoxin-PVIIA to Open and Closed Shaker K-Channels Are Differentially Affected by the Ionic Strength

**DOI:** 10.3390/md18110533

**Published:** 2020-10-26

**Authors:** David Naranjo, Ignacio Díaz-Franulic

**Affiliations:** 1Instituto de Neurociencia, Facultad de Ciencias, Universidad de Valparaíso, Valparaíso 2360103, Chile; 2Center for Bioinformatics and Integrative Biology, Facultad de Ciencias de la Vida, Universidad Andrés Bello, Santiago 8370146, Chile; ignacio.diaz@unab.cl; 3Centro Interdisciplinario de Neurociencia de Valparaíso, Facultad de Ciencias, Universidad de Valparaíso, Valparaíso 2360103, Chile

**Keywords:** Kv-channel, peptide toxin, conotoxin, association rate, dissociation rate, Brønsted-Bjerrum equation, predator-prey

## Abstract

κ-Conotoxin-PVIIA (κ-PVIIA) is a potassium-channel blocking peptide from the venom of the fish-hunting snail, *Conus purpurascens*, which is essential for quick prey’s excitotoxic immobilization. Binding of one κ-PVIIA to *Shaker* K-channels occludes the K^+^-conduction pore without additional conformational effects. Because this 27-residue toxin is +4-charged at neutral pH, we asked if electrostatic interactions play a role in binding. With Voltage-Clamp electrophysiology, we tested how ionic strength (IS) affects κ-PVIIA blockade to *Shaker*. When IS varied from ~0.06 to ~0.16 M, the dissociation constant for open and closed channels increased by ~5- and ~16-fold, respectively. While the association rates decreased equally, by ~4-fold, in open and closed channels, the dissociation rates increased 4–5-fold in closed channels but was IS-insensitive in open channels. To explain this differential IS-dependency, we propose that the bound κ-PVIIA wobbles, so that in open channels the intracellular environment, via ion-conduction pore, buffers the imposed IS-changes in the toxin-channel interface. A Brønsted-Bjerrum analysis on the rates predicts that if, instead of fish, the snail preyed on organisms with seawater-like lymph ionic composition, a severely harmless toxin, with >100-fold diminished affinity, would result. Thus, considerations of the native ionic environment are essential for conotoxins evaluation as pharmacological leads.

## 1. Introduction

Conotoxins form a family of peptides produced by marine predatory snails of genus *Conus* that bind to a wide variety of transporters, ion channels, G protein-coupled receptors and enzymes, providing a rich source of molecular templates and therapeutic leads for drug developing [1]. In fish hunting snails, a group of conotoxins acting in concert triggers an early excitotoxic shock; causing prey immobilization by axonal depolarization and repetitive firing near the venom injection lesion, followed by a long-term motor inhibition. The rapid elevated excitability arises because the venom blocks potassium channels and delays sodium channels inactivation [2]. κ-conotoxin-PVIIA (κ-PVIIA) is a 27-residue peptide forming part of the venom of *Conus purpurascens*, a piscivorous marine snail, that possibly is partially responsible for the excitotoxic shock in fish [2]. The toxin inhibition mechanism on voltage gated K-channels is well-known. As proposed for both scorpion α-KTx toxins and anemone toxins, κ-PVIIA inhibits voltage gated potassium channels by plugging the pore, thus, interrupting ion conduction [3,4,5]. Additionally, as proposed for the α-KTx chaybdotoxin (CTX), no induced conformational effects in the channel have been detected upon toxin binding [5,6,7]. Thus, the inhibition mechanism is the simplest possible; the toxin binding just occludes the pore. In addition, kinetic analyses of the toxin binding to potassium channels have revealed that κ-PVIIA blocks the pore with a very high association rate of 10–100 s^−1^μM^−1^, a value common among α-KTx toxins and in the range of diffusion limited protein associations [8,9]. Such mechanism of inhibition should be crucial for the early poisonous excitotoxic immobilizing shock of the prey, which is usually faster than the predatory snail [2].

As other animal toxins of the same family, κ-PVIIA is endowed with an excess of positive charges at its surface [10,11]. The spatial localization of positive charged residues in κ-PVIIA exhibits striking mimicry to that of CTX [10]. Thus, some of these charges may participate in the K-channel recognition surface. On the other hand, the potassium channel external pore entrance is dominated by negatively charged residues. Thus, both the molecular steering and recognition should be electrostatically facilitated [12]. To our knowledge, this hypothesis has not been tested. If so, this type of toxin would be largely ineffective for orthologous K-channels in marine animals having high ionic strength plasma, as some mollusks, echinoderms or arthropods.

In this work we study the effects of ionic strength in the kinetics of κ-PVIIA blockade on *Shaker* K-channels. We found that the association rate is very sensitive to the ionic strength, which is a result consistent with our hypothesis that the protein-protein recognition is mediated by through-space electrostatic interactions. Interestingly, the dissociation rate is also sensitive to the ionic strength in closed channels, but in open channels is ionic strength insensitive. We suggest that this lack of sensitivity in open channels is due to the fact that the toxin wobbles in the bound state [5].

## 2. Results

In normal amphibian plasma ionic strength, heterologous expression of voltage activated *Shaker* K-channels in *Xenopus* oocytes was detected as positive (outward) currents in response to positive-going voltage pulses between −80 and +50 mV from a holding potential of −90 mV (Figure 1). These voltage pulses drag the membrane potential to values more positive than the potassium equilibrium potential, thus they induce outward movement of K^+^-ions and, therefore positive currents. As the voltage pulse is made more positive, both the number of channels recruited and the speed with which they activate increase. Figure 1A–C shows current traces obtained from oocytes expressing *Shaker* K-channels in the absence and in the presence of 100 nM of κ-PVIIA perfused with three different recording solutions differing in ionic strength (50-Na^+^, 100-Na^+^ and 150-Na^+^; A, B and C, respectively; See methods). In the presence of κ-PVIIA (right traces), the activation kinetics appears delayed and the currents at the end of the voltage pulse slightly diminished in relation to their controls. As the ionic strength increases, the toxin induced inhibition gets weaker. Unfortunately, we had to limit the ionic strength excursion up to 0.16 M because higher ion concentrations induce osmotic stress that damages the oocytes and the quality of the recordings.

We have shown before that the apparent effect on the kinetics is due to the voltage dependence of the toxin/channel binding equilibrium [3,5,13]. Thus, 10 ms after the beginning of the voltage pulse, the observed time dependent relaxations are dominated by the transition from a high-affinity/binding equilibrium in closed channel, to a low-affinity, and voltage dependent, equilibrium in open channels. Figure 2 (left panel) shows these relaxations, obtained by computing point-by-point quotients between the traces with/without toxin from Figure 1, for different voltages (See methods).

All quotient traces start from approximately the same level of inhibition at the beginning of the voltage pulse (top of Figure 2A, left panel), which represents closed/resting channel inhibition, that later reach a different voltage dependent equilibrium inhibition after an exponential relaxation. Each trace was well described by a single exponential function that provided three parameters for each voltage pulse (blue traces): (a) the fraction of inhibited resting channels, which was estimated by extrapolating the fitting curve to the beginning of the voltage pulse (arrow; open symbols in Figure 2, middle panel), (b) the asymptotic fraction of inhibited open channels at each voltage (i_Tx_/i_Con_; filled circles in Figure 2, middle panel), and (c) the time constants for the voltage dependent relaxations (τ, Figure 2, right panel).

As the ionic strength increases, the inhibition of resting and open channels decreases. But the ionic strength effect on the inhibition at resting seems to be much more intense than that of the open state. Thus, for example, the resting fractional current rises from ~0.15 at 50-Na^+^ to ~0.7 at 150-Na^+^, which represents a ~14-fold rise in the dissociation constant, K_D_; from ~17 nM to 230 nM. Meanwhile, the fractional current at zero voltage (black symbols in Figure 2 middle panel) grows from 0.73 at 50-Na^+^ to 0.90 at 150-Na^+^ (a 3.4-fold rise in K_D_, from 260 to 900 nM). Thus, as expected, increased ionic strength decrease κ-PVIIA affinity for potassium channels, but interestingly, binding to a closed channel seems ~4-fold more sensitive to the ionic strength than it is to open channels. This difference could reveal a different electrostatic environment sensed by the toxin when interacting with open or closed channels. In order to address the mechanistic basis of such differential effects we determined the effect of ionic strength on the rate constants for toxin binding and unbinding in open and closed channels.

We and others have previously shown that a 1:1 stoichiometry describes adequately κ-PVIIA binding equilibrium [3,5,13,14]. We used the relaxation time constants (τ) and the voltage dependent steady-state inhibition (i_Tx_/i_Con_) in Figure 2 to determine, for each voltage, the association and dissociation rate constants, k_on_ and k_off_, respectively, according to the following equation system:(1)$$\tau =\frac{1}{k_{\mathrm{on}}\cdot \left[\mathrm{Tx}\right]+k_{\mathrm{off}}};\;\;\;\;\;(a)$$
$$\frac{i_{\mathrm{Tx}}}{i_{\mathrm{con}}}=\frac{k_{\mathrm{off}}}{k_{\mathrm{on}}\cdot \left[\mathrm{Tx}\right]+k_{\mathrm{off}}};\;\;(b)$$
where [Tx] is the toxin concentration. This two-unknown equation system was solved for k_on_ and k_off_ for each voltage. Figure 3 shows the result of this analysis on data from Figure 2 for 50-Na^+^, 100-Na^+^ and 150-Na^+^. As it happens with scorpion toxins having diffusion limited and electrostatically aided association rate to potassium channels [5,8,15,16], k_on_ is voltage independent, showing most of the ionic strength dependency, being at 150-Na^+^ 20–25% of that calculated for 50-Na^+^. On the other hand, the voltage dependent dissociation rate, k_off_ increases marginally at 150-Na^+^.

To obtain the kinetic constants of κ-PVIIA binding at resting, we used a two-pulse protocol devised by Terlau et al. [14] to test toxin rebinding to closed channels. This protocol consists of one conditioning voltage pulse that significantly relieves κ-PVIIA blockade, followed by a variable interval at the holding voltage to allow rebinding to closed channels. Then, a second (test) voltage pulse is given to activate channels and assess the amount of inhibition recovered. Figure 4 shows such protocol and the elements of the kinetic analysis of closed channels rebinding. As the interval gets longer, the toxin-induced inhibition gets stronger and the current traces begin to resemble those seen in the conditioning pulse, converging asymptotically to the closed channels binding equilibrium. By combining the time constants obtained from data in Figure 4 with steady state resting inhibition from Figure 2 (open symbols), with Equation system (1) we obtain k_on_ and k_off_ for the binding to the closed state. Figure 5 shows a summary of such analysis at different ionic strengths (actually, √IS). As k_on_ changes about the same ~5 fold for both, open and closed channels (black circles in Open and Closed panels), with the excursion from 50-Na^+^ to 150-Na^+^, the k_off_’s had opposite dependency on the ionic strength.

A decreasing association rate with the ionic strength is expected if the toxin and the channel are oppositely charged, thus, the attractive force between them is weakened by the interposed ionic cloud. As a simple quantitative analytical approach we used the Brønsted-Bjerrum equation to fit the kinetic data as function of the ionic strength. This equation predicts that the rate has an exponential dependency on the product of the charges and √IS (see Equation (5)). To do the fits, we fixed the κ-PVIIA charge, Z_A_ = +4e, according to the predicted peptide charge at neutral pH [10]. Thus, only two parameters were fit variables: the fitted vestibular charge, Z_B_, and the rate constant at zero IS, k_0_. For the association rates to open and to closed channels, Z_B_ was ~−2.5 e, suggesting that, regardless of the structural details, open and closed channels display similar electrostatic landscapes for the approaching κ-PVIIA. In contrast, unbinding rates from open and closed channels show a dramatic difference in their dependence on the ionic strength. On one hand, in open channels the IS-rise slightly slows unbinding, but in closed channels it conspicuously destabilizes the complex. Therefore, the fit parameter accounting for this difference, Z_B_, changes form −0.5 in open channels to ~+2.2 in closed ones, indicating different electrostatic surroundings for the bound toxin. We propose that the open channels reduced ionic strength sensitivity is suggestive that κ-PVIIA, as CTX, wobbles in the potassium channel mouth [5].

## 3. Discussion

We will discuss two ideas: (A) Unbinding of κ-PVIIA from open channels is less sensitive to changes in ionic strength than the unbinding from the closed ones. We suggest that because of toxin wobbling, the external mouth of the blocked pore equilibrates with the internal (unchanged) solution in the open channel, buffering the effect of the ionic strength of the external medium. In the closed channel such equilibration does not occur because the pore communication with the internal solution becomes interrupted by the closed channel activation gate [21]. (B) The electrostatically assisted binding of κ-PVIIA to Kv-channel suggests that the toxin could be far less effective if cone snails had predated on mollusks instead of fish.

The k_on_ for open and closed channels exhibits similar ~5-fold decrease from 50 to 150 mM of external NaCl (50-Na^+^ to 150-Na^+^), which, according the Brønsted-Bjerrum equation (Equation (5)), is expected for a bimolecular reaction between oppositely point-charged reactants having a charge-product, Z_A_ × Z_B_ ~−10e_o_^2^ (see legend for Figure 5) [22]. According to this model, k_on_ for κ-PVIIA would experience a 60–135-fold decrease when extrapolated to seawater ionic composition, in which the *C. purpurascens* lives. Similarly, the observed 4-fold increase in the dissociation rate from the closed channels is consistent with similar through-space toxin’s electrostatic attractive interactions for binding and unbinding of κ-PVIIA that are weakened by the ionic strength. Such effects have been reported for high affinity scorpion toxins [15,16]. Intriguingly, the smaller effect of the ionic strength on the dissociation rate from open channels suggests that the toxin bound feels significantly less-charged electrostatic surroundings in open channels. These effects have been observed with a low affinity CTX variant [16]. In fact, according to the simplistic approach of Equation (5), the charge difference between open and closed channels amounts to ~2.5 e_o_.

Functional and structural studies show that peptide scorpion toxins, as κ-PVIIA, bind to Kv-channels without functionally detectable conformational [7,13] or structural [6,23] effects. Thus, this differential dependency on ionic strength is intriguing. Two simple plausible scenarios come to mind to explain this disparity. One explanation could be that charge displacements during voltage activation change the electrostatic landscape. We know that voltage activation promotes a large volumetric and electrostatic remodeling because 12–16 charged arginine sidechains move outward, dragging hundreds of water molecules, in the voltage sensing domain [24,25,26,27]. Thus, the ~+2.5 extra charges could be a consequence of such large charge displacement. Nevertheless, such electrostatic remodeling should also be detected as a differential effect of the ionic strength on the association rate to open and closed channels. We did not see this (Figure 5), suggesting a more localized than global origin. Thus, we favor a local scenario in which κ-PVIIA wobbles while forming the binding complex [5]. Such activity partially detaches the toxin from its binding site allowing the external aspects of the pore to be connected with the internal solution. In the open channels, the k_off_ is dominated by the electrostatic effect coming from ions at the intracellular solution, which buffers the ionic strength in the vicinity of the toxin-channel interaction surface. Instead, when the channel that closes the communications with the intracellular solution is interrupted and unbinding depends solely on the ionic strength of the extracellular solution.

In an ecological context, such a large decrease in the dissociation constant would result in completely harmless toxin if the snail´s target were marine organisms with open circulatory systems (i.e., mollusks), whose lymph has similar ionic composition as sea water [20,28]. Instead, evolution drove the snail’s taste toward fish, in which the toxin-triggered excitotoxicity should be extremely effective due lower ionic composition in their blood. Figure 5 (right panel) suggests, according to the Brønsted-Bjerrum equation, that κ-PVIIA would have <1% of efficacy on identical Kv-channels if they were present in mollusks or crayfish relatives instead of fish. Thus, for assessment of conotoxins as therapeutic leads, a consideration of the native environment in which they function is essential for potency, selectivity, or pharmacokinetic evaluations.

## 4. Materials and Methods

Methods and reagent in general and κ-PVIIA are as in [3]. Briefly, stage V–VI oocytes were isolated from *Xenopus laevis* anesthetized by immersion in ice. Ovarian lobes were surgically removed and collected in ND96 solution (in mM): 96 NaCl, 2 KCl, 1.8 CaCl2, 1 MgCl2, 10 HEPES, pH 7.6 (50 μg/mL gentamicin, Aldrich Chemicals). Collagenase (Type II, Worthington) was used at a concentration of 0.9–1.5 mg/mL for digestion of tissue. After washing the enzyme, the oocytes were manually defolliculated in a nominally Ca^2+^-free ND96 solution.

The inactivation-removed *Shaker* K channel was expressed in *Xenopus* oocytes with in vitro transcribed cRNA injection (ranging from 50 to 200 pg per oocyte). The oocytes were incubated at 18 °C in ND96 supplemented with sodium pyruvate (2.5 mM) and BSA (0.04%) until subsequent recording. Macroscopic potassium currents were recorded with the two electrode voltage clamp techniques (TEVC) under continuous perfusion with a buffered saline containing (in mM): 96 NaCl, 2 KCl, 1 MgCl_2_, 0.3 CaCl_2_, 10 HEPES, and 25 μg/mL BSA, pH 7.6 (100-Na^+^). We used 0.3–1 MΩ microelectrodes, filled with 3M KCl, 5 mM EGTA and 10 mM HEPES (pH 7.0). The ionic strength was manipulated by adding extra 50 mM NaCl to the solution (150-Na^+^) or by replacing 50 NaCl with 100 mM mannitol in the solution (50-Na^+^). Recorded outward currents were evoked by voltage steps from –80 mV to +50 mV, from a holding potential of −90 mV (200 ms duration). Aliquots of κ-PVIIA were added into the bath solution immediately prior to perform the recordings. Calculations of the dissociation and association rate constants, k_off_ and k_on_, respectively, of κ-PVIIA blockade on open channels, were made from equation system (1) A concentration of 100 nM of toxin in the bath was used to measure both rate constants in open channels. The remaining fractional current at equilibrium (i_Tx_/i_con_) was used to obtain the dissociation constant, K_D_, with:(2)$$\frac{i_{\mathrm{Tx}}}{i_{\mathrm{con}}}=\frac{K_{D}}{K_{D}+\left[\mathrm{Tx}\right]},$$
where [Tx] is κ-PVIIA concentration. To measure kinetic constants of blockade on resting channels, 400 nM of κ-PVIIA were added to the solution and k_on_ was determined with:(3)$$k_{\mathrm{on}}=\frac{1}{\tau \cdot (K_{D}+\left[\mathrm{Tx}\right])}$$
in which τ is the time constant of blockade to closed/resting channels. Thus, from the above determinations, we finally define the entire system, obtaining the dissociation rate as:(4)$$k_{\mathrm{off}}=K_{D}k_{\mathrm{on}}$$

To predict the toxin binding equilibrium behavior in fish or mollusk plasma ionic condition we used the standard Brønsted-Bjerrum equation that evaluates the kinetic constants of a pseudo-first or second order reaction between ions [22].
(5)$$k=k_{o}e^{1.018\cdot Z_{A}\cdot Z_{B}\cdot \sqrt{I}S}$$
in which k_0_ is the kinetic constant at zero ionic strength, Z_A_ and Z_B_ are the ion´s electronic charge and IS, the ionic strength in M^1/2^. For all fits Z_A_ was fixed to +4 according to the theoretical κ-PVIIA charge [10] and only k_0_ and Z_B_ (the channel charge) were left as fitting variables. We calculated IS according to the following formula:(6)$$\mathrm{IS}=\frac{1}{2}\sum^{\;}C_{i}Z_{i}^{2}$$
in which C_i_ is the concentration (in mol/L) of ion i and Z_i_, its valence. All fits were made in Origin 6 (originlabs.com) with the Levenberg-Marquardt method.

## Figures and Tables

**Figure 1 marinedrugs-18-00533-f001:**
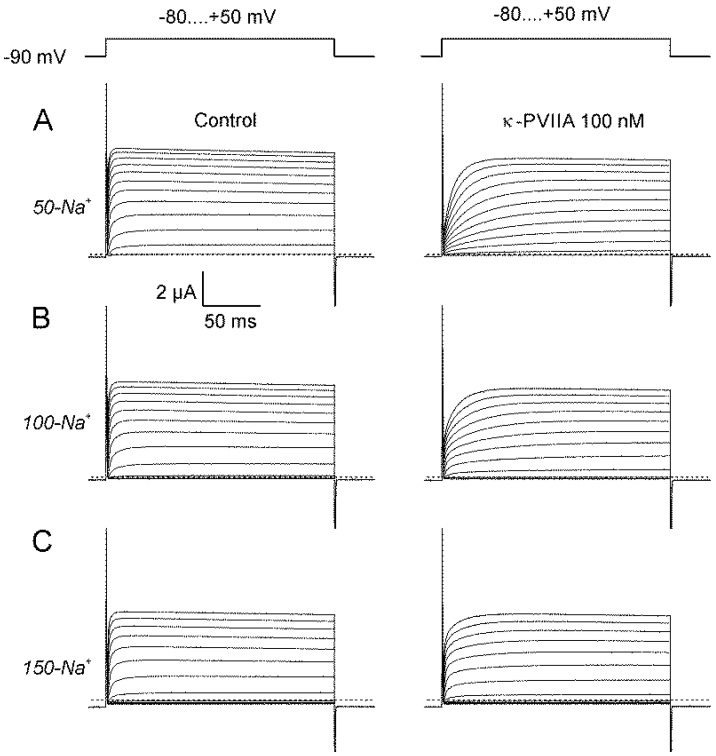
Ionic strength effects on the *Shaker* blockade by κ-Conotoxin-PVIIA (κ-PVIIA). Two-electrode voltage clamp traces records obtained with pulses between −80 and +50 mV with 10 mV increments from a holding voltage of −90 mV (top). External solutions in (**A**) 50-Na^+^; (**B**) 100-Na^+^; and (**C**) 150-Na^+^ were made changing the NaCl concentration but preserving all other ionic concentration. For 50-Na^+^, 100 mM mannitol was added to the solution for osmolarity compensation. Discontinuous lines indicate zero-current level.

**Figure 2 marinedrugs-18-00533-f002:**
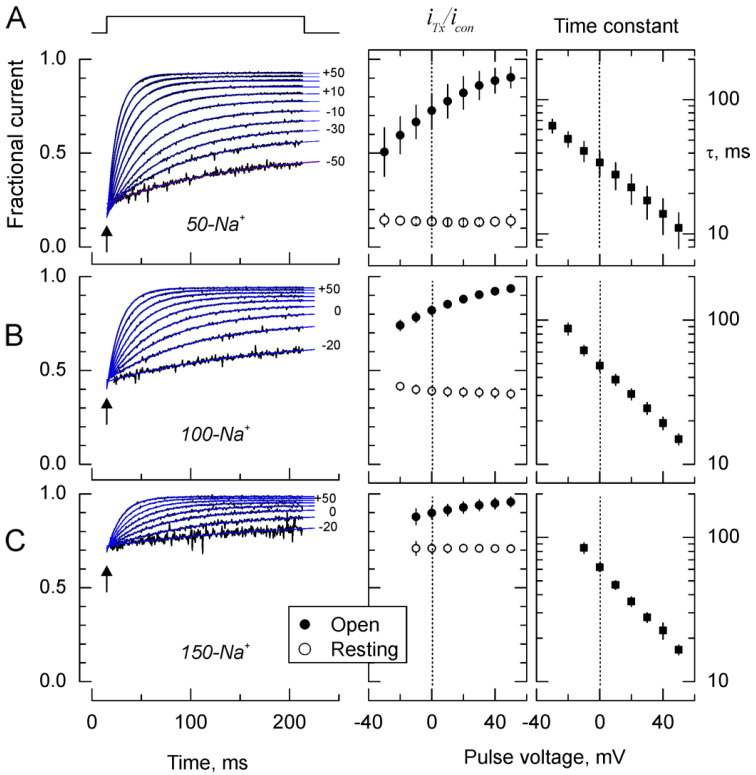
Kinetics of toxin blockade (left). Point by point quotients between records in the presence of 100 mM κ-PVIIA divided by records in its absence. (**A**) 50-Na^+^, (**B**) 100-Na^+^, and (**C**) 150-Na^+^ represent external solution containing 50, 100 and 150 mM NaCl, respectively. Traces represent the quotients measured from 5 ms after the beginning of the voltage pulse (vertical upward arrow) to 5 ms before its end. Traces were fitted to single exponential functions and extrapolated to the beginning of each voltage pulse (drawn in blue on top of the traces). Fractional steady state inhibition (Middle panel). Fractional steady state inhibition for open channels, obtained from the horizontal asymptotes of the fits, is represented in filled symbols (center). Steady state inhibition for closed channels, obtained from the extrapolation to the beginning of the pulse (upward arrows at left), is represented by open symbols (center). Time constants as function of the voltage for the fits (right). Each data point in the middle and left panels are mean ± SE for 4–6 individual determinations.

**Figure 3 marinedrugs-18-00533-f003:**
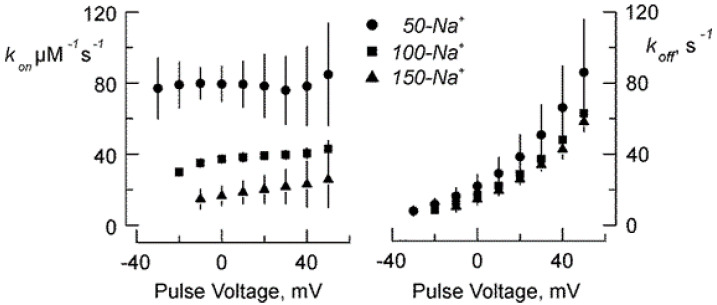
Rate constant as function of the voltage and ionic strength. Rate constants were calculated from the Equation system (1) with tau and steady state inhibition fits to data represented in the middle and right panels of Figure 2.

**Figure 4 marinedrugs-18-00533-f004:**
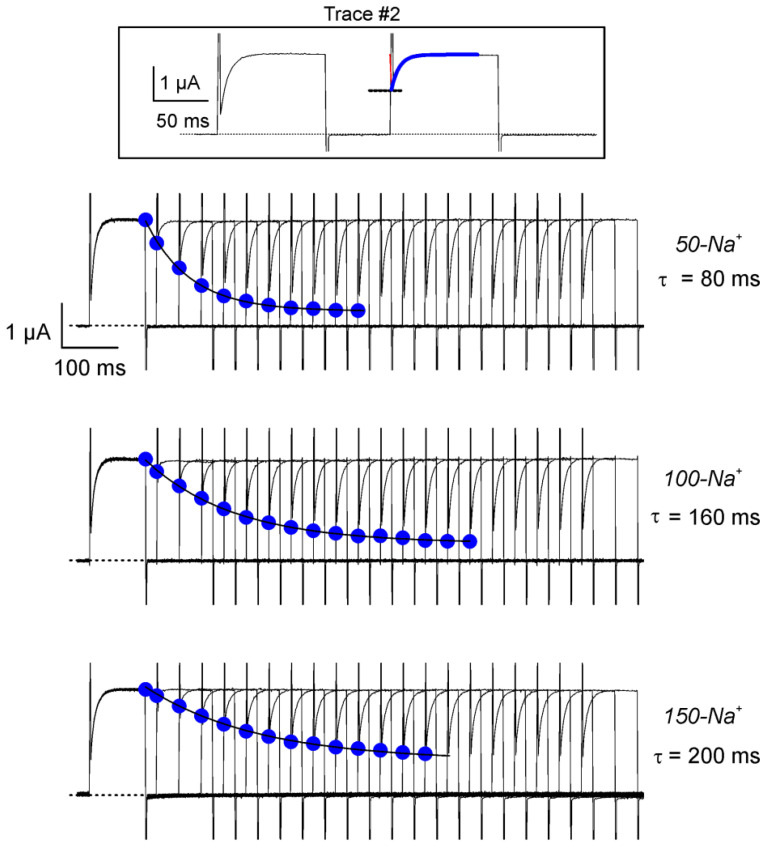
Two-pulse protocol to measure κ-PVIIA blockade kinetic to resting channels. A 100-ms pre-pulse to +50 mV was followed by a second 100-ms test pulse to +50 mV after a variable interval at −90 mV (Inset). The kinetics of re-inhibition at resting produced by 400 nM κ-PVIIA in the bath was estimated by extrapolating a single exponential fit to the beginning of the second pulse (thick blue trace intercept with dashed line). These inhibition values are plotted as filled blue circles along each set of traces. Then, the extrapolated data was fitted to a single exponential (blue traces) to estimate the time constant of toxin rebinding and the resting steady state inhibition.

**Figure 5 marinedrugs-18-00533-f005:**
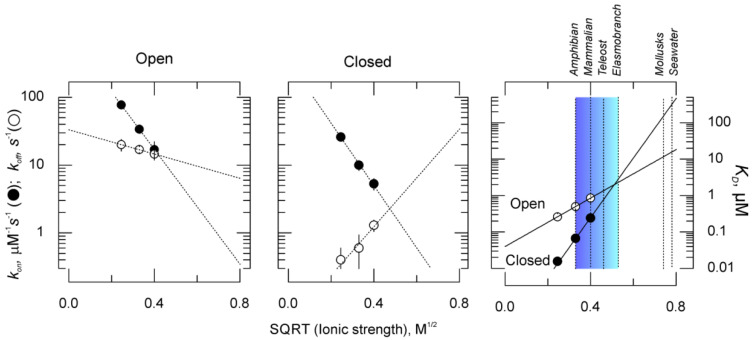
Summary of kinetic and equilibrium parameters of ionic strength effects on κ-PVIIA binding to *Shaker* K-channels. The rate and dissociation constants are plotted against the square root of the ionic strength. Rates were calculated from Equation (1) for open channel data at 0 mV, assuming 1:1 stoichiometry for open (left) and closed/resting channels (middle). Dashed lines are non-linear fits to the Brønsted-Bjerrum equation to define effect of the ionic strength on the rates (Equation (5)). For the fits, Z_B_ was fixed to +4 for all calculations, according to the theoretical κ-PVIIA charge at neutral pH [10]. Fit parameters for Open channels were: for k_on_; k_0_ = 835 ± 7 μM^−1^s^−1^ and Z_B_ = −2.4 ± 0.07; for k_off_: k_0_ = 33 ± 1 s^−1^ and Z_B_ = −0.5 ± 0.02. For closed channels k_on_: k_0_ = 364 ± 51 μM^−1^s^−1^ and Z_B_ = −2.6 ± 0.13; and k_off_: k_0_ = 0.036 ± 0.03 s^−1^ and Z_B_ = 2.19 ± 0.47. Dissociation constants (right), either experimental (symbols) or theoretical (lines), were calculated from Equation (4) show that closed channels exhibit stronger ionic strength dependency. Extrapolations based on the Brønsted-Bjerrum equation predict a 100–300-fold decrease in potency for resting channels if snail predated on mollusks instead of fish. Vertical dashed lines mark the approximate plasmatic ionic strength for amphibian, mammalian [17], estuarine and pelagic teleost fish [18], sharks [19] and mollusks [20].

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
