# Peer review of "Binding of κ-Conotoxin-PVIIA to Open and Closed Shaker K-Channels Are Differentially Affected by the Ionic Strength"

_marinedrugs, 2020, doi:10.3390/md18110533_

Round 1

Reviewer 1 Report

In this report, Naranjo and Diaz-Franulic examined the effect of ionic strength on the kinetics of the snail toxin (k-PVIIA)-mediated block of Shaker K+ channels heterologously expressed in Xenopus oocytes. The authors acquired K+ currents with 3 different external recording solutions: 50, 100 and 150 mM Na+. Under these conditions, the authors found that the affinity of k-PVIIA for the channel decreased as the ionic strength increased. Additionally, they found that the binding of the toxin was about 4-fold more sensitive to the ionic strength when the channel was closed than when it was opened. The authors employed the Bronsted-Bjerrum equation to determine both the kon and koff parameters under the three recording conditions.

The manuscript is well written, logically presented and clearly described. The authors have done an excellent job of describing their approach, their results and the physiological relevance of their findings. It was a real pleasure to read.

Author Response

We worked on the new version to, not only to answer the reviewer’s comments, but also to correct English and some format errors, homogenize the symbol nomenclature, and to improve its overall readability.  In consequence, we made no change that would affect the results, the interpretation, nor the conclusions of the manuscript

Reviewer 2 Report

The manuscript by Naranjo and Díaz-Franulic describes the analysis on the effect of ionic strength on the binding of the conotoxin κ-conotoxin-PVIIA. The approach is clear, the experiments were straight forward, and the results seems to support the authors conclusions.

One issue to be address is that on Figure 3 and related text, the authors show the rate of binding (Kon) is voltage independent. The authors attribute this to the fast that is a “diffusion-limited” rate. This conclusion would not be obvious for the readership and clearly for myself. The authors should expand on this issue.  

Author Response

In response to Referee #2, in the new version of the manuscript, we rewrote the text around Figure 3 to lower the emphasis that kappa-PVIIA association is diffusion limited. Our intention was to avoid mechanistic interpretations that are outside the scope of this work, and also because, to our knowledge, no measurements have been made to test for this type of mechanism in the kappa-PVIIA binding to Shaker K-channels. We just added the observation that the magnitude of the association rate is similar to that some scorpion toxins in which the diffusion limited character of the association have been tested. Thus,” As expected for a diffusion limited and electrostatically aided association rate….” was changed to: “As it happens with scorpion toxins having diffusion limited and electrostatically aided association rate … (Line 184 with the Track Changes-ON)